# Physicochemical, spectral, medicinal, and toxicological studies of Ketoprofen, Ibuprofen, and their major degradants: A quantum-chemical and in silico approach

**Protyoi Chakraborty**[1,2☯], **Saithajit Mohajan**[2,3☯], **Omme Samia**[2,4], **Nusrat Jahan**[2,3], **Nazmul Islam**[2,5], **Mahbub Alam**[6☯], **Monir Uzzaman**[2,7☯]*

1 Department of Chemistry, School of Physical Sciences, Shahjalal University of Science and Technology, Sylhet, Bangladesh, 2 Department of Drug Design, Computer in Chemistry and Medicine Laboratory, Dhaka, Bangladesh, 3 Department of Applied Chemistry and Chemical Engineering, University of Chittagong, Chattogram, Bangladesh, 4 Department of Pharmacy, Comilla University, Comilla, Bangladesh, 5 Department of Chemistry, Bhawal Badre Alam Government College, Gazipur, Bangladesh, 6 Department of Chemistry, Bangladesh Army University of Science and Technology Khulna, Khulna, Bangladesh, 7 Faculty of Science, Department of Chemistry, University of Chittagong, Chattogram, Bangladesh

☯ These authors contributed equally to this work.
* monircu92@gmail.com

## Abstract

Nonsteroidal Anti-inflammatory drugs (NSAIDs) are significantly consumed all over the world to treat pain, inflammation, and fever. Despite their ability to reduce pain and inflammation, they and their degradants can be toxic to both human organisms and the environment. This in-silico investigation has examined the spectral, physicochemical, biological, and toxicological properties of two frequently used NSAIDs, Ketoprofen (KTP) and Ibuprofen (IBP), along with some of their major degradants. Here, we have employed density functional theory, with the B3LYP/6-31g+(d,p) basis set, to determine the physicochemical as well as spectral properties of these compounds. Additionally, we have employed several computational methods to assess their biological and toxicological properties. We also executed molecular docking and nonbonding interaction calculations to investigate their binding properties and mode(s) of action against the Aspirin Acetylated Human Cyclooxygenase-2 receptor (PDB ID: 5F19), along with MD simulation of these complexes. 2-(4 methylphenyl) propanoic acid (IBP6), showed the greatest values of enthalpy and free energy, whereas 2-hydroxy Ibuprofen (IBP5) showed the highest binding interactions. For HOMO-LUMO values, 2-(3-(2-hydroxy-3-methylbenzoyl)phenyl) propanoic acid (KTP1) shows the lowest gap of 3.806 eV and 1-(4-isobutylphenyl) ethanol (IBP2) shows the highest gap of 6.063 eV. In case of binding affinity, 2-(3-(3- hydroxybenzoyl)phenyl) propanoic acid (KTP2) showed the highest value whereas, 3-hydroxy-carboxymethyl hydratopic acid (KTP3) and 2(-(4-formylphenyl) propanoic acid (IBP4) illustrated the lowest values of binding affinity. Additionally, ADMET and PASS predictions were evaluated to

**Data availability statement:** All relevant data are within the manuscript and its supporting information files.

**Funding:** The author(s) received no specific funding for this work.

**Competing interests:** The authors have declared that no competing interests exist.

compare their biological and toxicological parameters. The investigated complexes illustrated constancy over a 100-ns MD simulation, demonstrating crucial hydrogen bonding. Several of these degradants have demonstrated carcinogenic, nephrotoxic, and hematotoxic properties, which highlight their toxicological impact on human organisms. This investigative study can help raise awareness about the toxicological impacts of these drugs and their degradants on the environment as well as human health.

## 1. Introduction

NSAIDs are among the most widely used medications because they are effective at reducing pain and inflammation, which supports their inclusion in the WHO's model list of essential medicines. NSAIDs do more than just relieve pain, reduce inflammation, and lower fevers. They may also protect against serious diseases like cancer and heart disease [1]. They are commonly consumed for the temporary alleviation of moderate to severe pain, encompassing arthritis, musculoskeletal discomfort, dysmenorrhea, and cancers of the colon, breast, and prostate. They work by inhibiting COX enzymes, which lowers the synthesis of prostanoids and prostaglandins [2–4]. However, in recent years, due to incomplete removal in wastewater treatment and their potential to induce physiological effects at low doses, they are increasingly recognized as emerging pollutants [5]. These drugs can reach the environment through hospital or municipal wastewater, landfill leachate, solid waste management, or pharmaceutical effluents [6]. Given their persistent occurrence in water bodies and long-term toxic effects on living organisms, they are recognized as emerging environmental contaminants [5]. NSAIDs are among the most frequently occurring contaminants in the aquatic environment, mainly due to their high consumption, which ranks them among the most consumed medications [7–9]. In the environment, they have been found not only in the aquatic environment but also in soil, sediments, and even in drinking water. Their typical concentrations range from ng/L to µg/L. Despite these low concentrations, they can have toxic effects on living organisms based on their high bioactivity [10,11]. KTP, similar to IBP, has been reported to induce cardiac anomalies, immune system disruption, and genotoxic effects in aquatic organisms [12–14]. From the studies conducted so far, it can be said that IBP poses a high ecotoxicological risk, while KTP poses a moderate ecotoxicological risk to non-target organisms [14]. For this study, we have chosen KTP and IBP as they are among some of the most frequently used NSAIDs to reduce inflammation and manage mild-to-moderate pain [15–17]. However, the biological activity, molecular interaction potential, and toxicological implications of many of their reported degradation products remain insufficiently investigated. The degradants examined in this study were chosen because they have been repeatedly reported in the literature under a variety of environmentally relevant and treatment-related conditions. These compounds are often described as major and persistent transformation products and retain structural elements linked to biological activity. Their consistent detection under

realistic conditions underscores their environmental relevance, while the limited information available on their biological and toxicological effects provides a clear rationale for investigating them further in this work. For example, W. Song et al. reported an aerobic degradation of KTP through CA (Casamino Acids) consortium cultured with casamino acids that exhibited a higher KTP-degrading ability compared to those cultured with glucose, yeast extract, and mixed volumes; the proposed degradation pathway illustrated degradants such as 2-(3-(2-hydroxy-3-methylbenzoyl) phenyl) propanoic acid (KTP1), 2-(3-(3-hydroxybenzoyl) phenyl) propanoic acid (KTP2), and 3-(hydroxy-carboxymethyl) hydratopic acid (KTP3) [18]. J. Aguilar et al. investigated the photocatalytic degradation of ketoprofen with $TiO_2$ in aqueous media, reporting 3-benzoylbenzoic acid (KTP4) as an intermediate during the degradation process [19]. However, using a medium pressure (MP) lamp in pure water, filtered water, and all studies with 2-(3-(carboxy(hydroxy)-methyl)phenyl)propanoic acid (KTP5) as a degradant, R. Salgado et al. reported the evaluation of the photodegradation kinetics of KTP and a few other NSAIDs [20]. In addition to that, Feng, L., Oturan, N., van Hullebusch, E.D. et al. investigated the electrochemical degradation of KTP in tap water employing electro-Fenton (EF) and anodic oxidation (AO) processes with platinum (Pt) and boron-doped diamond (BDD) anodes and carbon felt cathode reporting degradants such as benzophenone (KTP6), 3-ethylbenzophenone (KTP7), 3-acetylbenzophenone (KTP8) and 2-[3-(hydroxy-phenyl-methyl) phenyl]propanoic acid (KTP9) [21] In the case of IBP, N. Sabri et al. evaluated the removal of IBP in the presence of hydrogen peroxide ($H_2O_2$) and sodium persulfate ($Na_2S_2O_8$) as oxidants activated by $Fe^{2+}$ where they identified 4-isobutylacetophenone (IBP1) and Oxo-Ibuprofen (IBP3) as major degradants [22]. Another study reported a degradation of IBP induced by hydroxyl radical, characterizing the intermediates through pulse radiolysis, identifying 1-(4-Isobutylphenyl)-1-ethanol (IBP2) as a degradant [23]. However, a degradation mechanism via UV/$H_2O_2$ process led to the formation of the degradant, IBP4 (2-(4 formyl-phenyl) propanoic acid) [24]. Another study revealed the formation of degradants, IBP5 (2-hydroxy Ibuprofen) and IBP6 (2-(4 methylphenyl) propanoic acid) via hydrodynamic cavitation of IBP [25].

Despite the identification of numerous degradants, their chemical reactivity, biological interaction potential, and toxicological implications remain insufficiently characterized. Since many degradants preserve pharmacologically relevant functional groups, they may retain the ability to interact with biological targets and exert biological or toxicological effects. Therefore, systematic investigation of their electronic properties, structural stability, protein-binding behavior, and predicted toxicity is essential for understanding their potential risks. In this study, an integrated computational approach was employed to investigate ketoprofen, ibuprofen, and their major degradation products using density functional theory, spectroscopic analysis, molecular docking, ADMET, PASS, $pIC_{50}$ studies, and molecular dynamics simulations. These methods were used to evaluate their physicochemical properties, electronic behavior, molecular interactions, and predicted biological and toxicological profiles. This study provides mechanistic insight into the behavior of NSAID degradants and highlights the importance of evaluating degradation products during pharmaceutical safety assessment and environmental risk evaluation.

## 2. Materials and methods

### 2.1. Geometry optimization

The structures of KTP and IBP were retrieved from the PubChem database. Full geometry optimizations were carried out using Gaussian 16, Revision C.01 [26]. Density functional theory (DFT), using B3LYP [27] hybrid functionals and Pople's 6-31G+ (d, p) basis set [28]. Molecular orbital parameters such as HOMO-LUMO energy gap (ΔE), chemical hardness (η), softness (S), and potentiality (μ) were computed using the following formulae [29];

$$\text{Gap } \Delta E = [\ \varepsilon\text{LUMO} - \ \varepsilon\text{HOMO}]$$

$$\eta = \frac{[\ \varepsilon\text{LUMO} - \ \varepsilon\text{HOMO}]}{2}$$

$$S = \frac{1}{2\eta}$$

$$\mu = \frac{[\varepsilon LUMO + \varepsilon HOMO]}{2}$$

## 2.2. FT-IR and UV spectral study

The FT-IR spectra of KTP, IBP, and their major degradants were theoretically simulated using Density Functional Theory (DFT)-based vibrational frequency calculations, employing B3LYP hybrid functionals. Therefore, no laboratory-based degradation or sample preparation was required to conduct the FT-IR analysis. Additionally, time-dependent density functional theory (TD-DFT) calculations were subsequently performed at the same level of theory (B3LYP/6-31g+ (d,p)) to study the electronic absorption (UV–Vis) spectra of the compounds [30].

## 2.3. Protein preparation, molecular docking, and interaction calculation

The binding affinity and mode(s) of KTP, IBP, and their degradants with the COX-2 target protein (PDB ID: 5F19) were investigated using molecular docking modeling [31]. The RCSB Protein Data Bank (PDB ID: 5F19) was utilized to collect the three-dimensional crystal structure of aspirin-acetylated human cyclooxygenase-2 (COX-2) in PDB format [32]. This structure was chosen because it has a high resolution of 2.04 Å [33] and is directly related to the inflammatory cyclooxygenase pathway. Crucially, the active binding site is precisely defined by the presence of a co-crystallized aspirin molecule, which improves the accuracy of docking analysis. The protein's chain A was utilized for docking investigations. There were no missing residues in the binding pocket region, and the structure matches that of Homo sapiens. Water, heteroatoms, and unexpected chains were removed using the Discovery Studio Visualizer 2021 program [34]. The PyMol software packages (version 4.6.0) were used to construct complexes of all the drugs and degradants with receptor proteins [35]. The Swiss-Pdb viewer software (version 4.1.0) minimized the protein's energy by removing poor protein atom connections while preserving the conjugate gradient approach [36]. The optimized drugs were tested using molecular docking against the human prostaglandin synthase protein (5F19). Finally, using PyRx (version 0.8) [37], a molecular docking simulation was carried out, treating the drugs and degradants as ligands and the protein as a macromolecule. Flexible docking was used in this analysis, converting all rotatable links into non-rotatable ones with center grid box sizes of 20.8612 Å, 37.5501 Å, and 59.3402 Å along the x, y, and z dimensions, respectively. Additionally, all of the docking findings were calculated, analyzed, and displayed using the Discovery Studio Visualizer 2021 program for non-bonding interactions [34].

## 2.4. ADMET, PASS, and drug-likeness prediction

Absorption, distribution, metabolism, excretion, and toxicity (ADMET) play a great role in drug design and toxicological analysis. Prediction of activity spectra for substances (PASS) elucidates the biological activity of organic compounds. 'Simplified molecular input line entry system (SMILES)' inputs were created with the assistance of the online server (https://cactus.nci.nih.gov/translate/) for the inputs of ADMET and PASS prediction. admetSAR (http://lmmd.ecust.edu.cn/admetsar1) has been used for ADMET prediction, and Way2Drug (http://www.way2drug.com/passonline/) online server for PASS prediction [29].

## 2.5. Molecular dynamics simulation

For a duration of 100 ns, molecular dynamics (MD) simulations were performed on a few of the degradants (KTP2, KTP9, IBP3, and IBP5), as well as the parent KTP and IBP complexed with the protein structure PDB ID: 5F19 derived from molecular docking investigations. Simulations using YASARA Dynamics [38] software with a time step of 2.50 fs were

conducted using the AMBER 14 force field. By utilizing the Berendsen thermostat, the systems were kept at a physiological pH of 7.4 and a temperature of 298 K [39]. The addition of $H_2O$ molecules with a 0.997 g/cm³ density and NaCl at a 0.9% concentration neutralized the systems. The Particle-Mesh Ewald (PME) approach was used to calculate electric interactions at long distances. With periodic boundary conditions, a cubic simulation cell was built, ensuring that its dimensions were at least 20 Å greater than those of the KTP-5F19 and IBP-5F19 complexes. Every 250 ps, MD trajectories were gathered for post-simulation examination.

### 2.6. Prediction of the quantitative structure-activity relationship (QSAR)

A comprehensive quantitative structure–activity relationship (QSAR) analysis was performed using eight molecular descriptors obtained from ChemDes (http://www.scbdd.com/chemdes/), an integrated web-based platform for molecular descriptor calculation, to illustrate the physicochemical and functional properties of the degradants [40].

## 3. Results and discussion

### 3.1. Thermodynamic analysis

Free energy and enthalpy, two of the major thermodynamic variables, are essential to understanding the possible stability, binding energy, and spontaneity of chemical reactions and their products [41,42]. The binding interactions with the receptor are intensified by negative free energy while decreased by positive free energy [43]. The geometrically optimized structures possess negative free energy and enthalpy values, meaning they release energy and therefore occur spontaneously, and form stable substances. Among all the degradants, IBP6 has the highest (least negative) enthalpy and free energy values, at – 538.594 Hartree and – 538.647 Hartree, and the lowest molecular weight. Among the KTP degradants, KTP6 has the highest (least negative) enthalpy and free energy values, at – 576.467 Hartree and – 576.516 Hartree. It is also the degradant with the lowest molecular weight, similar to IBP6 among the IBP degradants. Also, the lowest (most negative) values for both enthalpy and free energy were of KTP1 (- 994.059 Hartree) and IBP5 (- 222.284 Hartree), as illustrated in Figs 1 (a) and 2 (a), respectively. Both KTP6 and IBP6 exhibited the highest calculated enthalpy and Gibbs free energy values due to their lower molecular weights and reduced steric stabilization compared to other derivatives, and KTP1 and IBP5 elucidated the lowest values for these parameters because of their highest molecular weights and enhanced steric stability as a result of quantum chemical thermodynamic principles and large-scale benchmarks of gas-phase thermochemistry [44,45]. However, the IBP5 degradant illustrated the best binding interactions among all the other degradants, with a value of 6.459 Debye in terms of dipole moment. Dipole moment gives a signal of increased polarity which is a pivotal determining factor of binding affinities hence as per this investigation, IBP5 has the highest binding interactions in comparison other degradants and in the case of the degradants of KTP, KTP3 shows the highest determined value of binding interactions which is 5.122 Debye as illustrated in Figs 1 (b) and 2 (b) respectively. The presence of polar functional groups, hydroxyl (-OH) and carboxyl (-COOH) groups, is responsible for the large dipole moments of IBP5 and KTP3 that have been found. Large molecule dipole moments result from the considerable electronegativity discrepancies introduced by these groups. Dipole-dipole and hydrogen bonding interactions, which are essential for ligand-receptor binding and molecular recognition processes, are improved by such polarity [46,47].

### 3.2. Molecular orbital analysis

The transition from the ground state to the first excited state is linked to electronic absorption, which is primarily described by one electron excitation from HOMO to LUMO [48] that is also linked to chemical reactivity or stability. The HOMO-LUMO energy affects a molecule's electrophilic index, chemical potential values, softness, and hardness [3]. As the HOMO-LUMO gap increases, kinetic stability increases. As a result, moving electrons from the ground state HOMO to the excited state LUMO requires greater energy. According to this investigation, degradant IBP2 has the highest

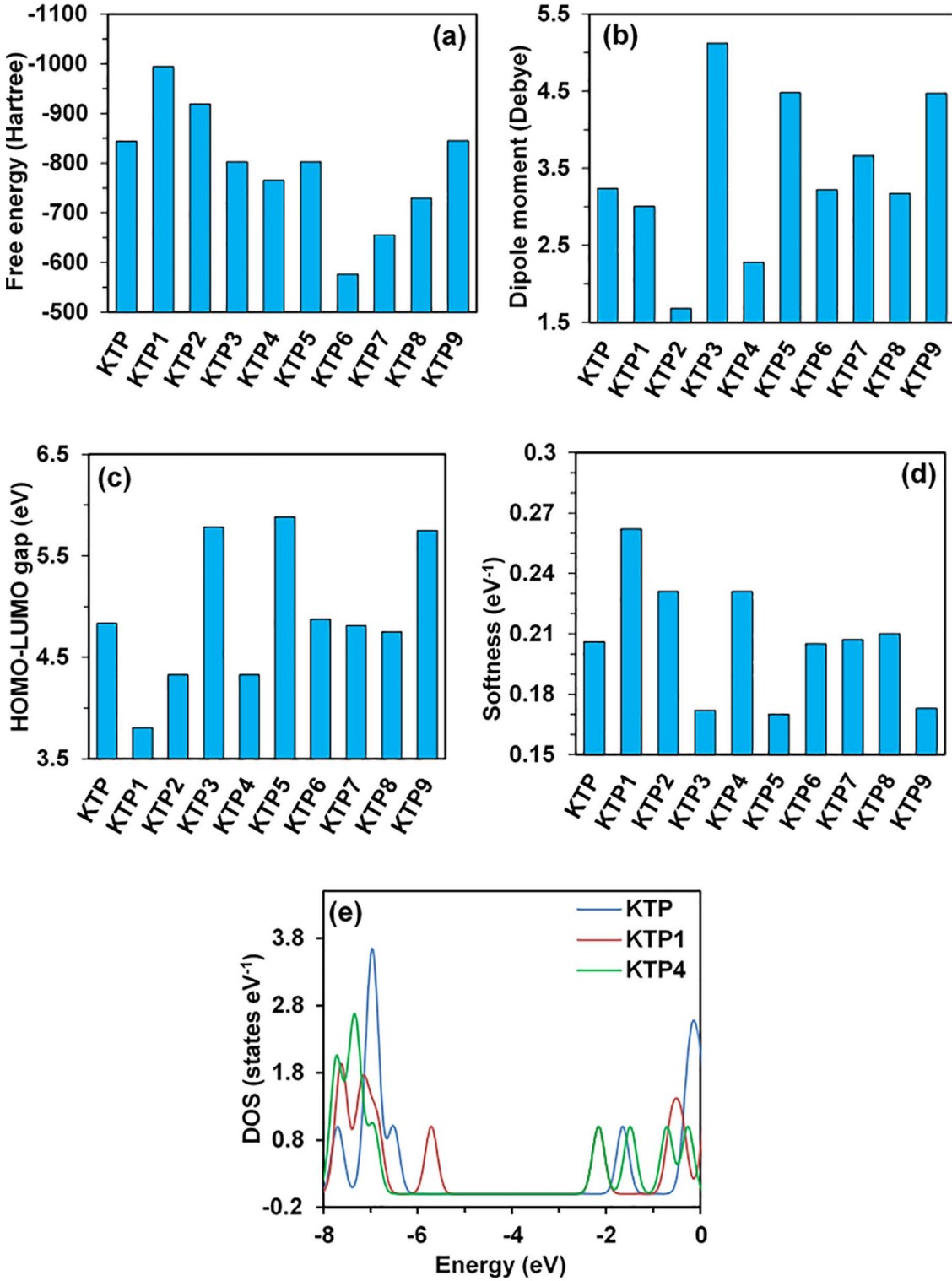

**Fig 1.** (a) Free energy, (b) dipole moment, (c) HOMO-LUMO energy gap, (d) chemical softness, and (e) DOS plots of KTP and its degradants.

HOMO-LUMO gap of 6.063 eV, likely due to delocalized electron density and reduced availability of frontier molecular orbitals for chemical interactions [49], with IBP2 has the highest HOMO-LUMO gap of 6.063 eV for less electron density and enhanced availability of molecular orbitals [49] as illustrated in Figs S2 (a) and S2 (b) in S1 File, IBP4 has the lowest chemical potential of − 4.781 eV, and IBP2 has the lowest chemical softness, which may have risen due to lower frontier orbital availability [49,50] as demonstrated in Figs 1(d) and 2(d). The lowest gap may suggest enhanced polarizability and chemical reactivity, while the highest gap suggests high kinetic stability and low chemical reactivity.

### 3.3. Electrostatic potential map analysis

Electrostatic potential (ESP) maps are indispensable tools used in molecular chemistry that illustrate the distribution of electric charges around a molecule. These maps are three-dimensional diagrams that highlight variably charged regions, aiding in the understanding of the spatial arrangement of charges within a molecule, which results in predicting the reactivity and interaction behavior of the molecule. The predictive power of docking algorithms can be improved by incorporating electrostatic potential map data into molecular docking simulations, allowing more accurate modeling of ligand-receptor interactions [51]. One can determine the number of positive or negative charges a molecule has by examining the sections of an ESP that display the tight packing in the region [52]. The electric density of each compound is displayed in red for polar and negatively charged regions, blue for non-polar and positively charged regions, and green for areas of zero potential, allowing the identification of probable sites for ionic interactions, hydrogen bonding, and other non-covalent forces [53,54] (Fig 3). As the chemical potential is inversely correlated with the charge transfer, the charge transfer property is significantly influenced by the chemical potential value. Investigated compounds that bear the highest positive and negative potentiality also exhibit the best possible index of nucleophilic and electrophilic attack, respectively. Among the KTP, IBP, and the degradants, the two largest positive potential values were found for KTP4 (+ 0.216 Hartree) and IBP5 (+ 0.180 Hartree) for each case, which indicate a powerful nucleophilic characteristic that may increase their reactivity toward electrophilic sites. Meanwhile, the highest negative potential was found for KTP7 (- 0.250 Hartree) and IBP1 (- 0.244 Hartree), which favors interactions with nucleophilic targets because of their electrophilic nature. All remaining figures have been presented in supporting information in the Fig S4 in S1 File.

### 3.4. FT-IR analysis

Fourier Transform Infrared Spectroscopy (FT-IR) is a widely employed analytical method for characterizing and identifying the chemical composition of materials by analyzing their infrared absorption spectra. Each material produces a unique spectrum, often described as its 'chemical fingerprint' [55]. In this study, frequencies ranging from 400 to 4000 $cm^{-1}$ were calculated and adjusted using a scaling factor of 0.9648 to enhance accuracy [56]. For KTP and its nine degradants, aromatic C–H stretching peaks appeared between 3086–3104 $cm^{-1}$, while linear C–H stretching was observed at 2929–2950 $cm^{-1}$. Aromatic C=C vibrations were indicated by multiple adjacent peaks in the 1569–1593 $cm^{-1}$ range. The presence of the C=O bond was confirmed in almost all the compounds as they exhibited vibrational frequency in between 1611−1777 $cm^{-1}$. The highest C=O stretching was exhibited by KTP9 (1777 $cm^{-1}$), and the lowest one was by KTP1 (1611 $cm^{-1}$). O-H bonds have been discovered in several compounds with a frequency range of 3591−3691 $cm^{-1}$, except for KTP1 (3121 $cm^{-1}$).

Meanwhile, stretching movements at the range of 3058−3091 $cm^{-1}$ were confirmed for IBP and its six degradants. IBP and its degradants show linear C–H stretching vibrations in the range of 2922–2947 $cm^{-1}$. Aromatic C=C bond vibrations for IBP and its derivatives were observed between 1559–1602 $cm^{-1}$. The C=O bond was detected in nearly all compounds, except for IBP-2, with vibrational frequencies spanning 1610–1776 $cm^{-1}$. O-H bonds have been discovered in several compounds with a frequency range of 3121−3691 $cm^{-1}$. The complete spectrum is illustrated in Fig 4(a) and 4(b), Figs S5 (a) and S6 (a) in S1 File, alongside the data presented in Tables S3 (a) and S3 (b) in the supporting information S1 File.

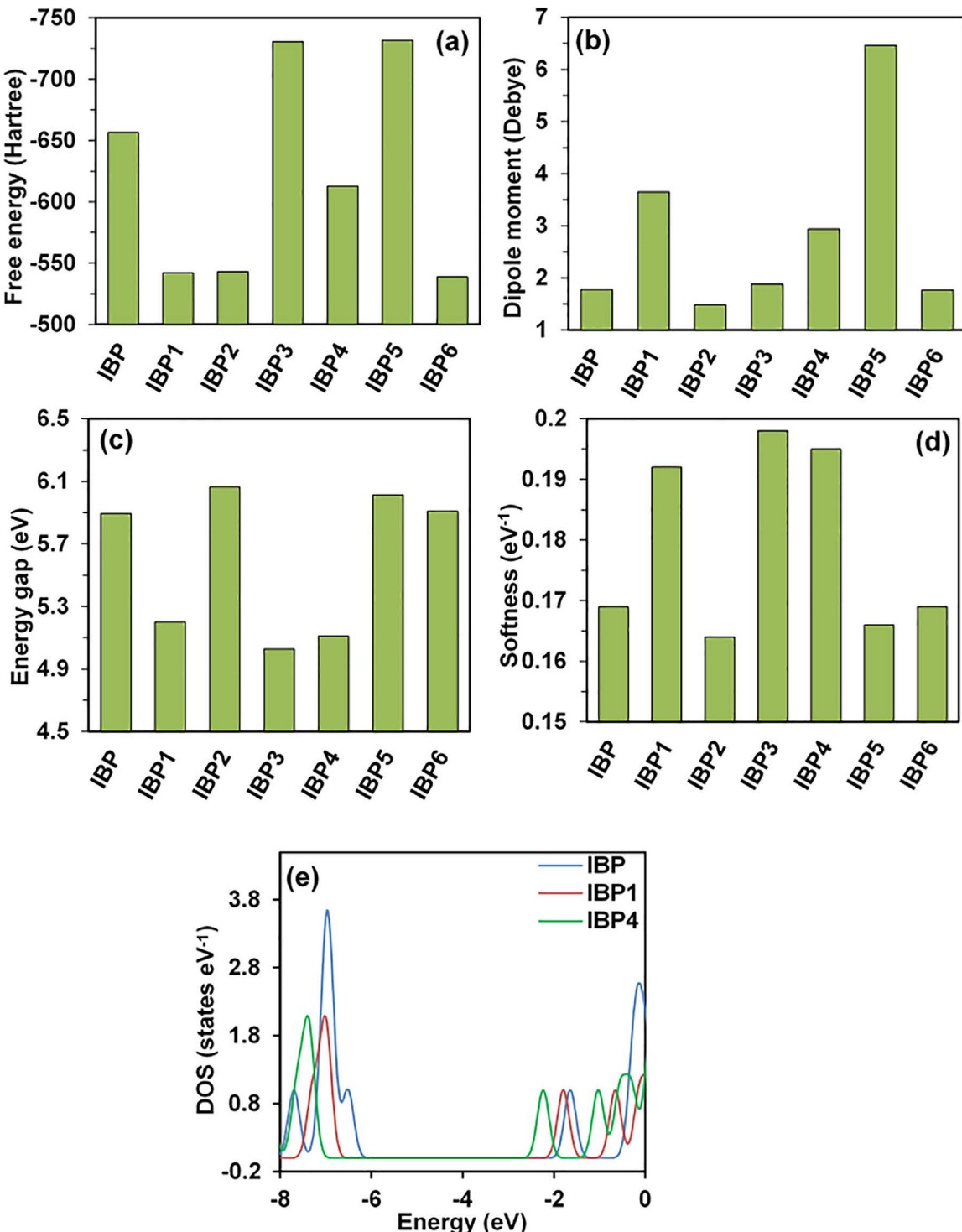

**Fig 2.** (a) Free energy, (b) dipole moment, (c) HOMO-LUMO energy gap, (d) chemical softness, and (e) DOS plots of IBP and its degradants.

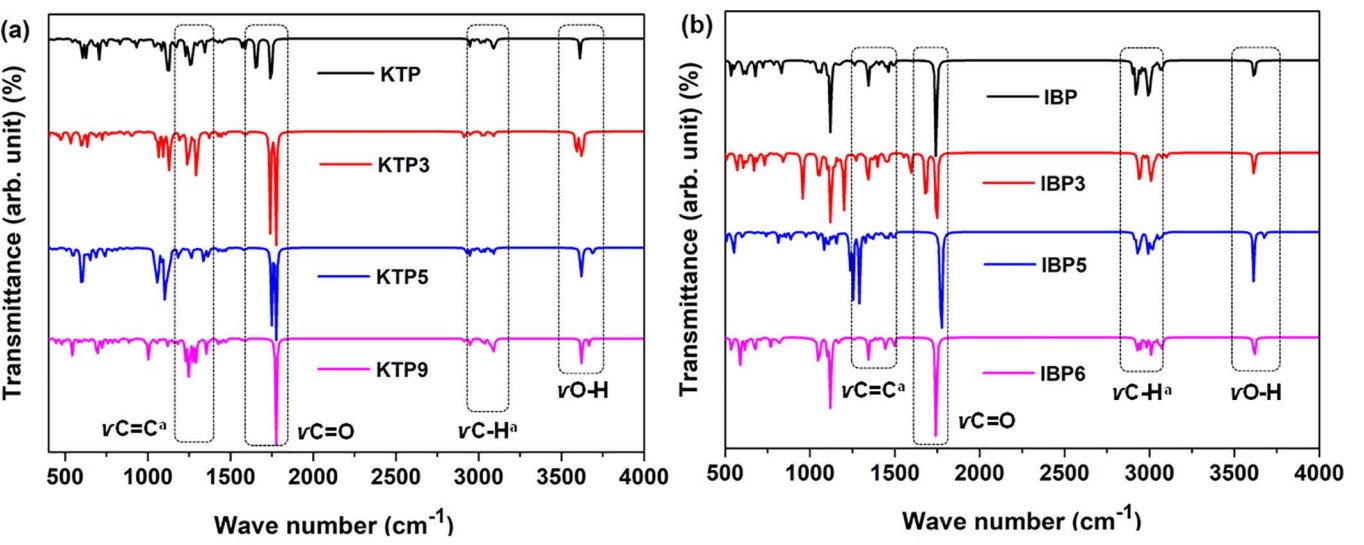

**Fig 3. ESP map of (a) KTP, (b) IBP, and some of their degradants, illustrating the distribution of electron density.**

**Fig 4. FT-IR spectra of (a) KTP, (b) IBP, and their degradants.**

## 3.5. UV-Visible spectral analysis

UV–Visible spectroscopy offers a reliable approach for evaluating electronic absorption or emission through time-dependent density functional theory (TD-DFT). Its combination of accuracy and computational efficiency makes it highly effective for predicting electronic absorption spectra, photophysical behavior, and charge-transfer dynamics [57]. In this study, the initial electronic transition from the ground state ($S_0$) to the first singlet excited state ($S_1$) dictates both kinetic stability and chemical reactivity.

This study provides comprehensive insights into the excitation energies, wavelengths, and oscillator strengths of IBP, KTP, and their degradants, emphasizing their influence on reactivity, stability, and absorption properties. Excitation energy, reflecting the HOMO–LUMO gap, is critical in balancing stability and reactivity: higher excitation energies, corresponding to larger gaps, enhance kinetic stability but reduce chemical reactivity, whereas lower energies, associated with smaller gaps, increase reactivity at the expense of stability [58].

For IBP and its degradants, the compounds with higher excitation energies, such as IBP (4.242 eV), IBP2 (5.076 eV), IBP5 (4.264 eV), and IBP6 (4.249 eV), are predicted to be more kinetically stable and less reactive. Among these, IBP2 exhibits the highest excitation energy (5.076 eV), indicating the largest HOMO–LUMO gap and therefore the highest predicted stability. On the other hand, IBP1 (3.524 eV), IBP3 (3.370 eV), and IBP4 (3.450 eV) exhibit lower excitation energies, which are linked to higher reactivity and lower stability [58]. The oscillator strengths for IBP and its degradants range from 0 to 0.061.

Similarly, KTP and some of its degradants with lower excitation energies, such as KTP (3.166 eV), KTP1 (2.953 eV), KTP2 (3.130 eV), KTP4 (3.131 eV), KTP6 (3.163 eV), KTP7 (3.156 eV), and KTP8 (3.135 eV), exhibit higher reactivity and lower stability. In contrast, KTP9 (4.234 eV), KTP5 (4.222 eV), and KTP3 (3.972 eV) show higher excitation energies, indicating greater stability and reduced reactivity. The oscillator strengths for KTP and its degradants range from 0 to 0.027.

Overall, the results suggest that the degradants of KTP exhibit lower excitation energies, indicating higher reactivity but lower stability than those of IBP. Among the degradants, IBP2 exhibits the highest stability, evidenced by its higher excitation energy. Together with the data shown in Tables S4 (a) and S4 (b) in the supporting information S1 File, the whole spectrum is depicted in Fig 5(a) and 5(b), Figs S5 (b) and S6 (b) in S1 File.

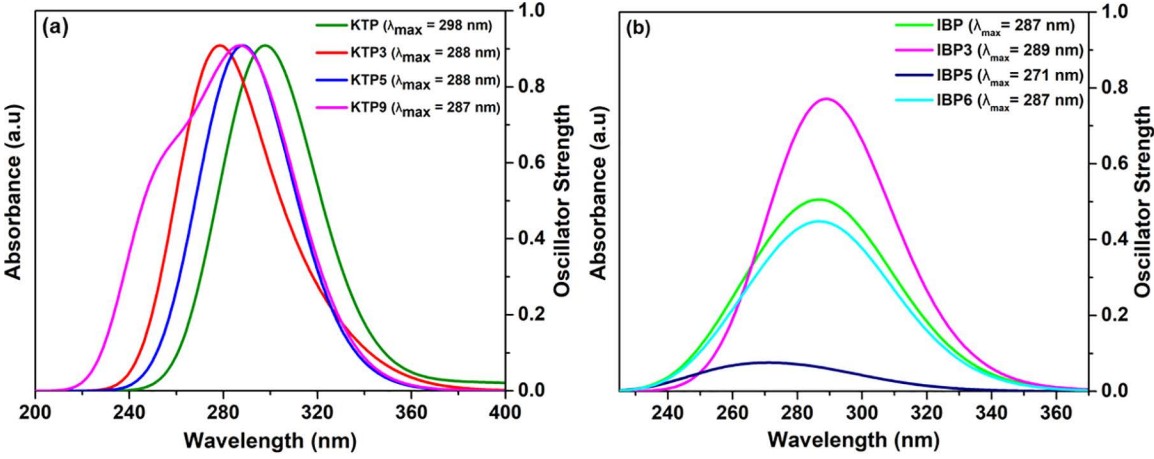

**Fig 5. UV-Vis spectra (normalized) of (a) KTP, (b) IBP, and their degradants.**

### 3.6. Analysis of molecular docking and non-bonding interactions

Molecular docking, which examines interactions among multiple molecules to form stable complexes, plays a crucial role in applications such as hit identification, lead optimization, bioremediation, and rational drug design. Electrostatic potential maps help determine how ligands bind by offering information about the potential properties and charge distribution of receptor proteins [29]. The discovery of new bioactive chemicals can be greatly improved by combining computational biomolecular docking with creative laboratory approaches [59]. To determine the ideal ligand conformations, molecular docking simulations are frequently used. These simulations estimate the orientation of ligands and the binding affinity at the receptor protein's active site while taking the system's total energy into account [35,60].

This study centered on the aspirin-acetylated human cyclooxygenase-2 receptor (PDB ID: 5F19), a protein targeted by aspirin and other NSAIDs to inhibit prostaglandin synthesis [32]. The main objective of our docking study was to assess the binding affinities of KTP, IBP, and their degradation products as prostaglandin inhibitors, along with their compatibility with the selected protein. The parent drugs in this study are KTP and IBP. Fig 6 (a) shows that the parent drug KTP has a binding affinity of – 8.27 kcal/mol, which is higher than the degradants KTP3 (- 6.97 kcal/mol), KTP5 (- 7.30 kcal/mol), KTP6 (- 7.53 kcal/mol), KTP7 (- 7.73 kcal/mol), KTP8 (- 8.07 kcal/mol), and KTP9 (- 8.13 kcal/mol). With binding affinities ranging from – 6.97 kcal/mol to – 9.47 kcal/mol across all derivatives, the degradants KTP1 (- 8.80 kcal/mol), KTP2 (- 9.47 kcal/mol), and KTP4 (- 8.57 kcal/mol), however, outperform the parent molecule. The parent drug, IBP, exhibits a binding affinity of – 7.23 kcal/mol in Fig 6 (b), but its degradants have a binding affinity of −6.97 kcal/mol to −7.93 kcal/mol. IBP4 has the lowest affinity (- 6.97 kcal/mol), followed by IBP2 (- 7.07 kcal/mol), IBP1 and IBP6 (both – 7.20 kcal/mol), IBP5 (- 7.40 kcal/mol), and IBP3 (- 7.93 kcal/mol), which has the highest affinity among the degradants of IBP. This molecular docking approach resembles the lock-and-key model [61], where a more negative binding energy value signifies a stronger interplay between the receptor proteins and the ligands [62].

The binding interactions of KTP and IBP, including their degradants, with the 5F19 receptor protein are presented in Figs 6 (a) and (b). In addition, the hydrogen surface area of protein 5F19 for certain compounds is shown in Figs 7 (a) and 7 (b), where pink regions indicate donors (electronegative atoms with a lower electron density) and green regions indicate hydrogen bond acceptors (electronegative atoms with a higher electron density). These illustrations highlight the critical function of hydrogen bonds in utilizing complementary acceptor/donor interactions to stabilize non-covalent interactions [63]. Stronger hydrogen bonds (less than 2.3 Å) greatly increase binding strength, which is a major factor in increased ligand-receptor

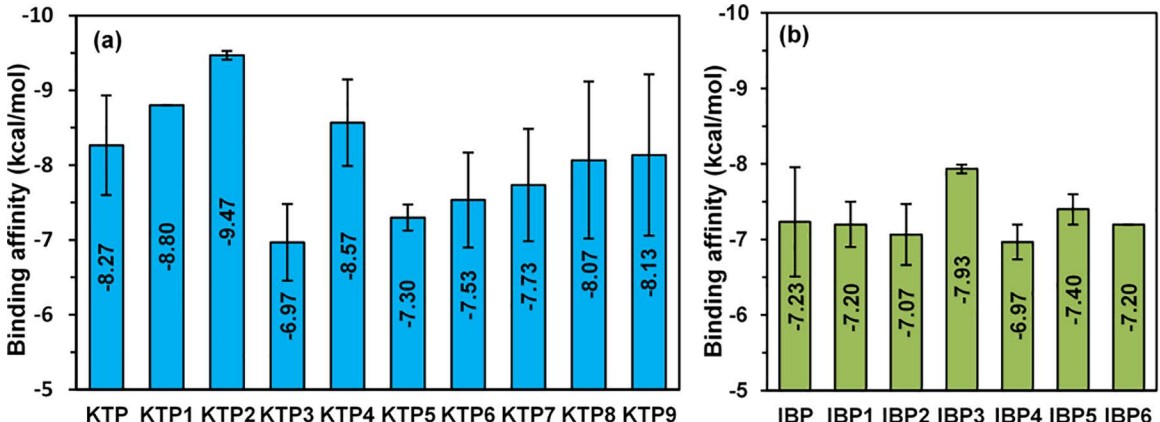

**Fig 6. Comparative binding affinity of (a) KTP, (b) IBP, and some of their degradants with COX-1(PDB ID: 5F19), illustrating their selectivity and potential for targeted COX-2 inhibition.**

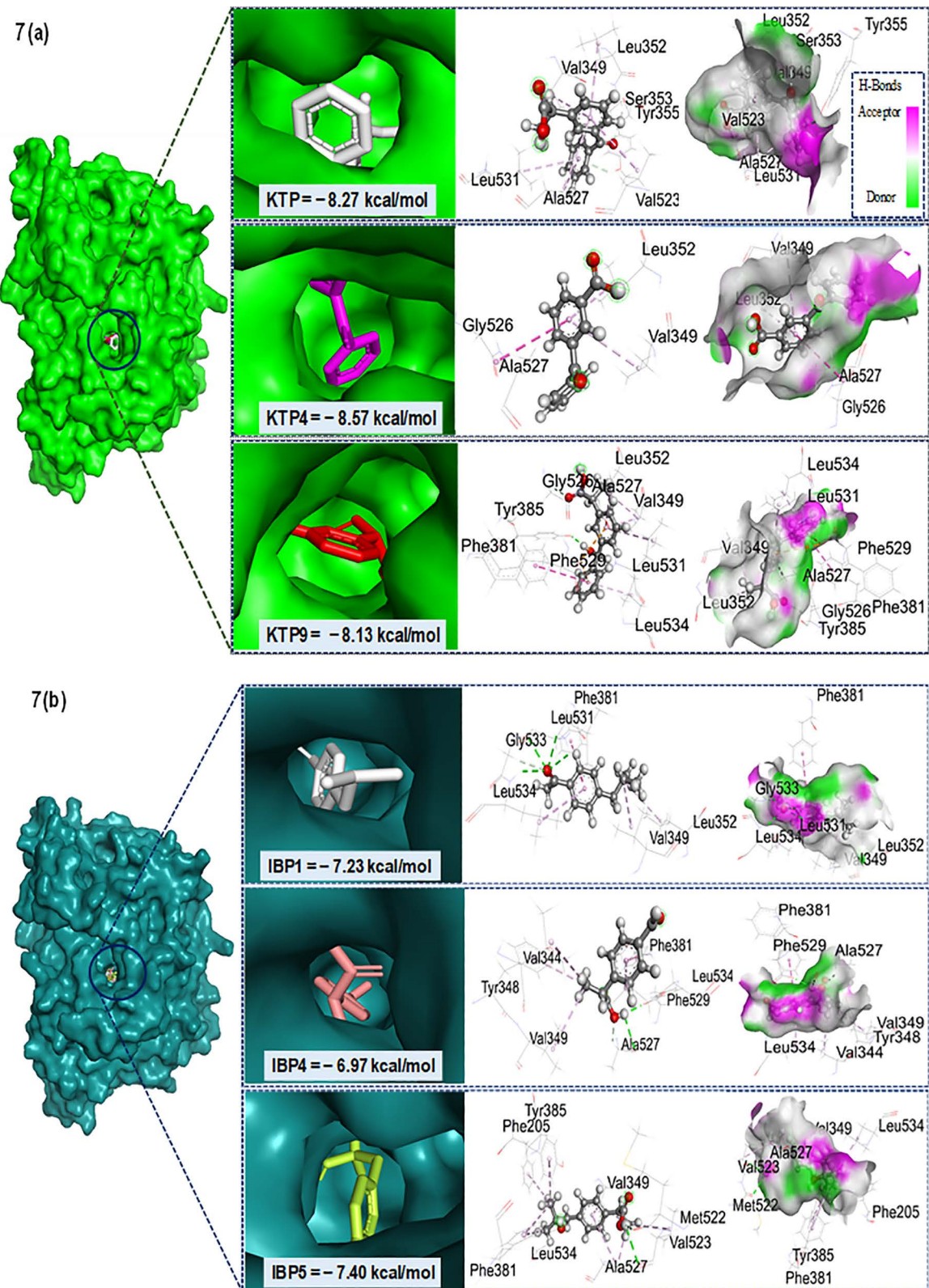

**Fig 7. Docked conformers, non-bonding interactions, and hydrogen bond surfaces of some of (a) KTP and (b) IBP's degradants with receptor protein 5F19.**

binding affinity [64]. Furthermore, non-covalent interactions are critical components of the ligand-protein complex, since they stabilize the drugs at their target site, thus enhancing their efficacy and modifying their binding affinity [47]. All examined complexes with the 5F19 protein were stabilized through non-covalent interactions such as hydrogen bonding and hydrophobic contacts. Nearly all compounds formed conventional hydrogen bonds as well as pi–alkyl interactions with key residues. KTP1, KTP2, and KTP4 primarily formed pi-alkyl bonds and conventional hydrogen bonds with common residues VAL349, GLY526, and ALA526 at different distances (for KTP2 and KTP4) and with ARG44, TYR130, and CYS41 residues at 1.828 Å, 2.087 Å, and 1.961 Å distances, respectively (for KTP1). Their high binding scores and efficacy are largely due to these hydrogens, hydrophobic, and van der Waals interactions and bond distances [65,66]. However, some other weak carbon-hydrogen bonds with SER353 (at KTP), ARG44 (at KTP1), GLY526, GLY533 (at KTP2), GLY526 (at KTP3), HIS39, ARG469, GLU465 (at KTP5), PRO156 (at KTP8), GLY (at KTP9), HIS39, PRO40 (at IBP), GLY533 (at IBP1), TYR385 (at IBP3), and ALA526 (at IBP4) residues were found responsible for a slight conformational change due to larger distances (>2.3Å) [65]. As hydrogen bond length increases, the interaction energy generally decreases; such weak hydrogen bonding can slightly alter ligand orientation and consequently induce minor structural adjustments within the binding pocket [67]. Besides, π-systems improve ligand-protein recognition by considerably contributing to binding energy via π-bond interactions in biological systems [41,68]. Some important residues were found with Alkyl and Pi-Alkyl bonds in the maximum compounds. Notably, KTP3, KTP7, and KTP8 created Pi-Donor Hydrogen bonds with TYR355, TYR130, and CYS47, but KTP3, KTP9, and IBP6 only formed Pi-Anion bonds with PHE529. Several types, such as Pi-Pi-stacked, Pi-Sigma, Pi-Pi-T-shaped, and Amide-Pi stacked interactions, were also identified in some of the degradants within the 5F19 protein, demonstrating the structural diversity of ligand-protein binding (see Tables S5 (a) and S5 (b)) in S1 File.

Ultimately, docking analysis revealed that KTP and the majority of its degradants have a higher binding affinity than IBP and the majority of its degradants. KTP1, KTP2 (highest binding affinity, − 9.47 kcal/mol), KTP4, KTP8, and KTP9 have developed a comparatively greater relationship with the receptor protein among KTP, IBP, and their degradants.

### 3.7. ADMET analysis

ADMET features are essential for predicting the pharmacokinetic properties of KTP, IBP, and their degradants to assess their toxicological effects [1]. Pharmacokinetic and toxicological parameters have been selected from ADMET results, which are indicated in Tables 1 and 2. ADMET is the abbreviation of 'absorption, distribution, metabolism, excretion, and

Table 1. Selected pharmacokinetic parameters of KTP and its degradants.

| Name | Absorption | | Distribution | | | Metabolism | Toxicity | | | | |
|---|---|---|---|---|---|---|---|---|---|---|---|
| | HIA+ | C2P+ | BBB+ | P-GpI | P-GpS | CYP450 2C9 | hERG | Carcinogen | BID | AOT | FT |
| KTP | 0.993 | 0.887 | 0.938 | NI(0.959) | 0.713 | NI(0.907) | WI(0.966) | NC(0.630) | 0.670 | II | 0.888 |
| KTP1 | 0.906 | −0.641 | −0.527 | NI(0.976) | 0.525 | NI(0.655) | WI(0.976) | NC (0.885) | 0.555 | III | 0.991 |
| KTP2 | 0.962 | 0.710 | 0.604 | NI(0.972) | 0.599 | NI(0.596) | WI(0.953) | NC(0.822) | 0.515 | III | 0.963 |
| KTP3 | 0.873 | −0.652 | 0.683 | NI(0.979) | 0.630 | NI(0.855) | WI(0.986) | NC(0.768) | 0.849 | III | 0.883 |
| KTP4 | 0.988 | 0.883 | 0.947 | NI(0.965) | 0.777 | NI(0.956) | WI(0.971) | NC(0.662) | 0.784 | IV | 0.885 |
| KTP5 | 0.873 | −0.652 | 0.683 | NI(0.979) | 0.630 | NI(0.855) | WI(0.986) | NC(0.768) | 0.849 | III | 0.883 |
| KTP6 | 0.997 | 0.939 | 0.984 | NI(0.949) | 0.774 | NI(0.879) | WI(0.925) | NC(0.616) | 0.611 | IV | 0.797 |
| KTP7 | 1.000 | 0.928 | 0.980 | NI(0.853) | 0.664 | NI(0.721) | WI(0.885) | 0.521 | 0.594 | III | 0.875 |
| KTP8 | 1.000 | 0.916 | 0.980 | NI(0.898) | 0.726 | NI(0.709) | WI(0.903) | NC(0.605) | 0.680 | III | 0.738 |
| KTP9 | 0.994 | 0.881 | 0.809 | NI(0.965) | 0.655 | NI(0.768) | WI(0.976) | NC(0.617) | 0.555 | IV | 0.815 |

HIA = Human Intestinal Absorption, C2P = Caco-2 permeability, BBB = Blood Brain Barrier, P-GpI = P-Glycoprotein Inhibitor, P-GpS = P-Glycoprotein Substrate, hERG = Human ether a-go-go-gene, AOT = Acute Oral Toxicity. FT = Fish toxicity, BID = Biodegradation, NC = Non-carcinogens.

**Table 2. Selected pharmacokinetic parameters of IBP and its degradants.**

|  | Absorption | | Distribution | | | Metabolism | Toxicity | | | | |
|---|---|---|---|---|---|---|---|---|---|---|---|
| Name | HIA+ | C2P+ | BBB+ | P-GpI | P-GpS | CYP4502C9 | hERG | Carcinogen | BID | AOT | FT |
| IBP | 0.993 | 0.887 | 0.962 | NI(0.932) | 0.759 | NI(0.931) | WI(0.972) | 0.555 | 0.514 | III | 0.947 |
| IBP1 | 1.000 | 0.909 | 0.983 | NI(0.816) | 0.714 | NI(0.933) | WI(0.924) | 0.557 | 0.559 | III | 0.888 |
| IBP2 | 0.996 | 0.899 | 0.975 | NI(0.878) | 0.731 | NI(0.911) | WI(0.909) | 0.590 | 0.616 | III | 0.847 |
| IBP3 | 0.991 | 0.821 | 0.909 | NI(0.958) | 0.738 | NI(0.903) | WI(0.980) | 0.500 | 0.702 | II | 0.841 |
| IBP4 | 0.988 | 0.824 | 0.934 | NI(0.992) | 0.766 | NI(0.981) | WI(0.974) | NC(0.617) | 0.796 | II | 0.853 |
| IBP5 | 0.989 | 0.736 | 0.822 | NI(0.909) | 0.535 | NI(0.801) | WI(0.990) | NC(0.547) | 0.870 | III | 0.777 |
| IBP6 | 0.992 | 0.919 | 0.954 | NI(0.992) | 0.801 | NI(0.971) | WI(0.975) | NC(0.519) | 0.771 | III | 0.721 |

HIA = Human Intestinal Absorption, C2P = Caco-2 permeability, BBB = Blood Brain Barrier, P-GpI = P-Glycoprotein Inhibitor, P-GpS = P-Glycoprotein Substrate, hERG = Human ether a-go-go-gene, AOT = Acute Oral Toxicity. FT = Fish toxicity, BID = Biodegradation, NC = Non-carcinogens

toxicity', which aids in predicting pharmacokinetic characteristics and toxicological analysis [69]. In Table 1, KTP and its degradants demonstrated positive outcomes except for KTP1, KTP3, and KTP5, and in Table 2, IBP and its degradants demonstrated positive results. Degradants of KTP and IBP are generally found in waste products of hospitals and industries (drug-production factories), which create acute adverse effects in our environment because they disrupt the ecosystem by mixing up in food chains [70]. Table 1 and 2 illustrate that KTP, IBP, and their degradants, especially IBP1 (1.000), KTP7 (1.000), and KTP8 (1.000), are absorbed well into human intestinal absorption (HIA), and the rest of the compounds result in more than 0.90, except KTP3 and KTP5. Higher absorption depends on higher Caco-2 permeability [4]. In Tables 1,2, the majority of compounds have higher C2P+ values of more than 0.80 without IBP5, KTP1, KTP3, and KTP5. Also, drug distribution means that medications are distributed in different parts of the body. It also has various criteria, such as how well it crosses the blood-brain barrier (BBB), how well it works as a P-glycoprotein inhibitor, how well it gets into the central nervous system (CNS), and how well it moves through the P-glycoprotein substrates [71]. All drugs and their degradants were predicted to cross the blood–brain barrier (BBB), except KTP1, suggesting permeability through active transport mechanisms and/or transcellular passive diffusion. P-glycoprotein has great value in pharmacology, drug absorption, drug distribution, drug elimination, and drug interaction. For example, in pharmacology, P-glycoprotein is named as a multidrug antagonist protein [72]. In metabolism, CYP450 enzyme inhibitors help in biotransformation [73]. All drugs and degradants demonstrate weak inhibitors (WI) of human ether a-go-go-related gene (hERG), which means they are quite safe for the cardiac muscles [74]. Among the degradants of KTP, all were predicted to be non-carcinogenic except KTP7. In contrast, IBP and its derivatives IBP1, IBP2, and IBP3 exhibited carcinogenic potential with values ranging from 0.500 to 0.600, while IBP4, IBP5, and IBP6 were classified as non-carcinogenic. Moreover, all these drugs and degradants show III acute oral toxicity, where IBP3, IBP4, and KTP show II acute oral toxicity, KTP4, KTP6, and KTP9 show IV acute oral toxicity. Overall, all drugs and their degradants demonstrated fish toxicity (FT) more than 0.700 in ranged from 0.721 to 0.991, where KTP1 had the highest value. HIA depends on pKa and log $P_o$/w [75]. HIA value is different because pKa and log $P_o$/w of all degradants are not the same. Except for KTP7, KTP and its degradants are non-carcinogenic due to their cytotoxic and anti-cancer actions on ovarian cancer, inhibition of cyclooxygenase, and reduction of tumor growth [76]. IBP biodegradation processes produce hydroxylated and carboxylated degradants, which are thought to be more toxic than IBP itself [70]. A BOILED-Egg model has been generated using SwissADME software to evaluate human intestinal absorption (HIA) and blood-brain barrier (BBB) permeability of KTP, IBP and their respective degradants. KTP and its degradants exhibited high HIA, as all compounds were located within the white region of the BOILED-Egg plot Fig 8(a) However, variations in BBB permeability were observed, with KTP1 and KTP3 showing limited penetration, likely due to their higher polarity (increased topological polar surface area, TPSA) and lower lipophilicity,

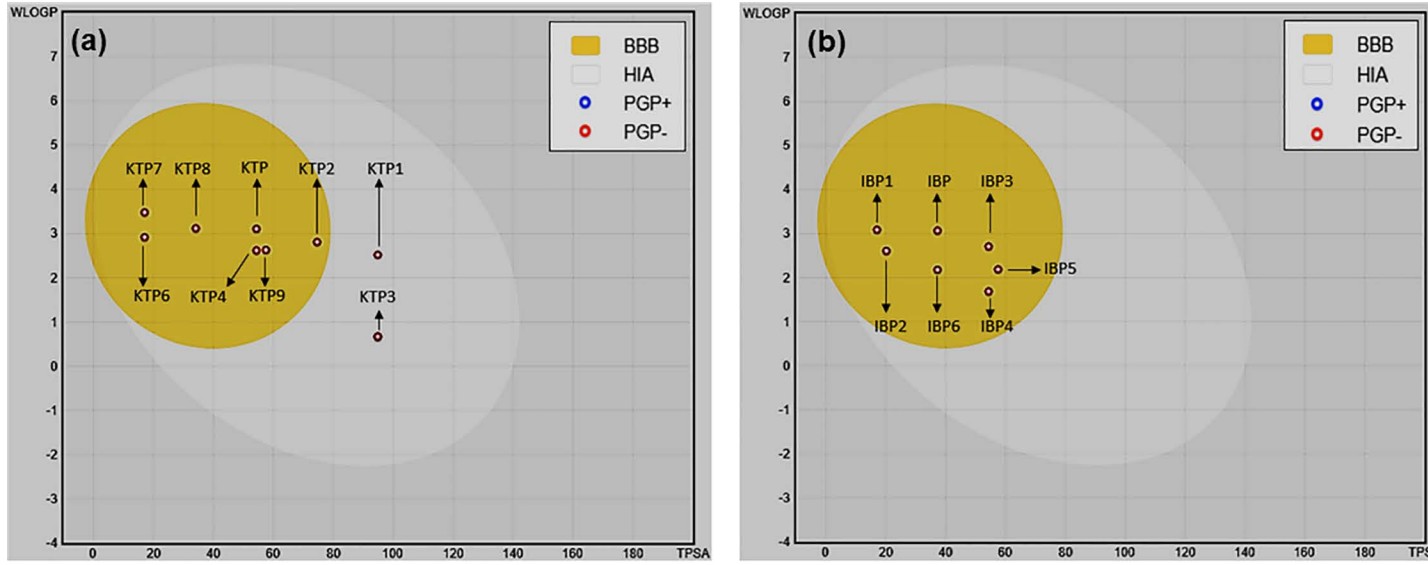

**Fig 8. BOILED-Egg diagram showing gastrointestinal absorption and the blood-brain barrier (BBB) permeability of (a) KTP, (b) IBP and their degradants.**

which restrict passive diffusion across biological membranes [77]. IBP and its degradants demonstrated consistently favorable pharmacokinetic properties, with all compounds showing high HIA and most positioned within the BBB-permeable region of the BOILED-Egg plot Fig 8(b). This behavior can be attributed to their lower TPSA values, moderate lipophilicity, and reduced efflux potential, indicate better absorption and distribution characteristics compared to KTP derivatives [77].

### 3.8. PASS prediction

Pharmacological effects, such as analgesic, antipyretic, and anti-inflammatory, and toxicological effects, such as gastrointestinal hemorrhage, necrosis, nephrotoxic, hematotoxic, and hepatitis, biochemical mechanisms are represented in 'Prediction of activity spectra for substances (PASS)' [78]. The results of PASS vary from 0.00 to 1.00. From Tables 3 and 4, pharmacological effects (analgesic) of maximum degradants $P_a < 0.550$ and antipyretic range of Pa value from 0.281 to 0.936, whereas antipyretic $P_a$ value of KTP, KTP1, KTP2, KTP4, KTP9, IBP, IBP3, IBP4 and IBP6 are more than 0.700; rest of the degradants are less than 0.700. Besides, IBP2 seems to have the lowest (0.348) anti-inflammatory $P_a$ value, and KTP indicates the maximum (0.925) anti-inflammatory $P_a$ value, except KTP4, KTP7, KTP8, IBP1, and IBP2 rest of which have anti-inflammatory $P_a$ values more than 0.700, which are more preferable. KTP, IBP, and all of their degradants exhibited high $P_a$ values for gastrointestinal hemorrhage in the range of 0.761 to 0.964. KTP2 showed the highest (0.964) gastrointestinal hemorrhage $P_a$ value, and KTP8 showed the lowest (0.761) gastrointestinal hemorrhage Pa value. The majority of the degradants demonstrated $P_a$ values of necrosis toxicity more than 0.700, except KTP6 ($P_a = 0.610$), KTP7 ($P_a = 0.593$), and KTP8 ($P_a = 0.545$). Furthermore, possible hepatitis ($P_a$) values for KTP7, KTP8, IBP1, IBP2, IBP4, and IBP5 are less than 0.700, and the rest of the substances are more than 0.700. Nephrotoxic data of all substances for medications in the range from $0.423 < P_a < 0.775$, KTP2 is more nephrotoxic because of a higher value than other degradants. Hematotoxic data of all degradants range from $0.426 < P_a < 0.786$; KTP2 shows a higher hematotoxic effect than the others. KTP and IBP exhibit antipyretic, analgesic, and anti-inflammatory effects by reversibly slowing down COX-1 and COX-2 enzymes, thereby reducing the production of proinflammatory precursors [79,80]. The S-isomer of the racemic

**Table 3. PASS prediction of the KTP and its degradants.**

| Name | Analgesic | Antipyretic | Anti-inflammatory | GH | Necrosis | Hepatitis | Nephrotoxic | Hematotoxic |
|------|-----------|-------------|-------------------|-----|----------|-----------|-------------|-------------|
| KTP | 0.549 | 0.898 | 0.925 | 0.953 | 0.802 | 0.864 | 0.726 | 0.729 |
| KTP1 | 0.369 | 0.936 | 0.903 | 0.934 | 0.823 | 0.805 | 0.738 | 0.702 |
| KTP2 | 0.490 | 0.934 | 0.905 | 0.964 | 0.857 | 0.892 | 0.775 | 0.786 |
| KTP3 | 0.250 | 0.570 | 0.839 | 0.895 | 0.829 | 0.774 | 0.680 | 0.723 |
| KTP4 | 0.293 | 0.722 | 0.652 | 0.895 | 0.731 | 0.771 | 0.603 | 0.574 |
| KTP5 | 0.250 | 0.570 | 0.839 | 0.895 | 0.829 | 0.774 | 0.680 | 0.723 |
| KTP6 | 0.380 | 0.487 | 0.703 | 0.867 | 0.610 | 0.864 | 0.726 | 0.460 |
| KTP7 | 0.275 | 0.502 | 0.524 | 0.890 | 0.593 | 0.666 | 0.423 | 0.426 |
| KTP8 | 0.234 | 0.588 | 0.662 | 0.761 | 0.545 | 0.636 | 0.483 | 0.441 |
| KTP9 | 0.385 | 0.750 | 0.907 | 0.909 | 0.786 | 0.783 | 0.706 | 0.738 |

**Table 4. PASS Prediction of IBP and its degradants.**

| Name | Analgesic | Antipyretic | Anti-inflammatory | GH | Necrosis | Hepatitis | Nephrotoxic | Hematotoxic |
|------|-----------|-------------|-------------------|-----|----------|-----------|-------------|-------------|
| IBP | 0.463 | 0.818 | 0.901 | 0.891 | 0.820 | 0.817 | 0.745 | 0.734 |
| IBP1 | 0.286 | 0.514 | 0.465 | 0.856 | 0.603 | 0.624 | 0.526 | 0.458 |
| IBP2 | 0.296 | 0.281 | 0.348 | 0.854 | 0.750 | 0.669 | 0.597 | 0.601 |
| IBP3 | 0.363 | 0.917 | 0.909 | 0.931 | 0.768 | 0.743 | 0.652 | 0.657 |
| IBP4 | 0.242 | 0.754 | 0.863 | 0.907 | 0.718 | 0.624 | 0.569 | 0.496 |
| IBP5 | 0.332 | 0.537 | 0.874 | 0.926 | 0.704 | 0.696 | 0.667 | 0.657 |
| IBP6 | 0.538 | 0.796 | 0.924 | 0.950 | 0.845 | 0.857 | 0.760 | 0.783 |

Gastrointestinal hemorrhage = GH.

combination KTP is the only one with cyclo-oxygenase-inhibiting activity; the R-isomer is way less effective [81]. Hemato-toxicity leads to severe health issues, such as anemia, polycythemia, leukopenia, and leukemia [82].

### 3.9. Molecular dynamics simulations

Molecular dynamics simulation (MDS) is a key method in modern drug design that lets researchers study how biomolecules interact with each other and how the structure of a receptor affects its biological function. To verify the molecular docking results of our study, 100 ns MD simulations were performed on two analogs (KTP2, KTP9) along with the parent molecule KTP and two analogs (IBP3, IBP5) along with IBP employing the YASARA dynamics package. Trajectory analysis were used to look at how stable and dynamic the complexes were, including root mean square deviation (RMSD), radius of gyration (Rg), and root mean square fluctuation (RMSF). We assessed the RMSD values of C-alpha atoms from the docked complexes using the simulated trajectories to get a better idea of how the ligand–protein distance changes and how the structure maintains conformational stability over time. The stable RMSD profiles of all these compounds are displayed in Figs 9 (a) and 9 (b). In contrast to KTP, which displayed a value of 1.914 ± 0.265 Å, the mean RMSD values for KTP2 and KTP9 were 1.729 ± 0.242 Å and 1.833 ± 0.269 Å. However, IBP3 and IBP5 displayed RMSD values of 2.099 ± 0.363 Å and 1.952 ± 0.319 Å, which were similar to the RMSD value of the parent IBP, which was 1.787 ± 0.230 Å. The stable RMSD profiles suggest that there are no significant conformational changes and demonstrate a strong interaction between ligands and the active site [83]. As shown in Figs 9 (a) and 9 (b), the average hydrogen bond values for the 5F19 protein complexed with KTP, KTP2, and KTP9 are 432.378, 432.030, and 435.446, respectively. Furthermore, IBP,

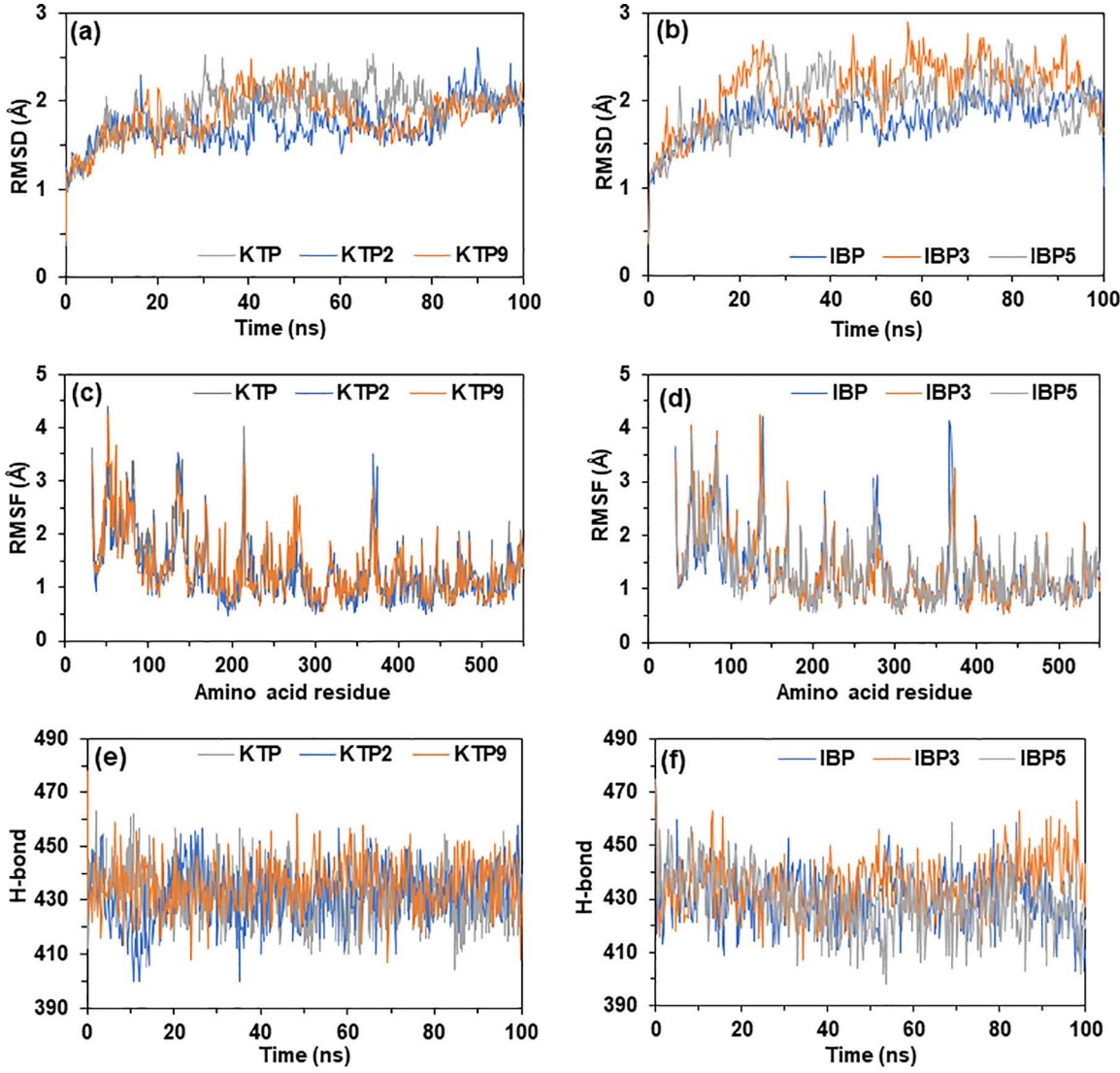

**Fig 9. MD simulations over 100 ns: Root mean square deviation (RMSD) (a) KTP, (b) IBP; Root mean square fluctuation (RMSF) (c) KTP, (d) IBP; Hydrogen bonds (e) KTP, (f) IBP; Radius of gyration (Rg).**

IBP3, and IBP5 reported scores for the hydrogen bonds of 430.913, 435.397, and 428.996, respectively. IBP3 showed the best hydrogen bond profile, which indicates higher ligand stability [83]. For the Rg values of KTP at around 55 ns and KTP2 at about 95 ns, there are two minor increasing fluctuations. KTP9 reported the lowest mean value of Rg, 24.436, indicating that this analog has increased rigidity and compactness in 5F19 as demonstrated in Figs S8 (a) and S8 (b) in S1 File. For each of the six simulated complexes, the RMSF based on a single residue was quite comparable. This investigation demonstrates that all three analogs show little fluctuation during the 100 ns MD simulation and are stable in their initial binding site as elucidated in Figs 9 (a) and 9 (b). When the analogs are attached to the protein, the MD simulation study demonstrates that they are reasonably stable [84]. Furthermore, KTP9 showed the highest Solvent Accessible Surface Area (SASA) value of 24839 among all the analogs, indicating that KTP9 has a larger surface area exposed to the solvent, which suggests it is not buried well in the binding site, and KTP with a SASA value of 24035 suggests the most

favorable SASA profile as illustrated in Figs S8 (a) and S8 (b) in S1 File [85]. The average data of all these profiles of the analogs have been presented in Table 5.

### 3.10. pIC$_{50}$ studies

The Quantitative Structure–Activity Relationship (QSAR) technique offers a strong computational foundation for predicting the biological activity of various compounds. In our study, a validated multiple linear regression (MLR) model has been employed for establishing a correlation between eight molecular descriptors: Chiv5, bcutm1, MRVSA6, MRVSA9, PEOEVSA5, GATSv4, J, and Diametert; [86] and the biological activity endpoint pIC$_{50}$, which is the negative logarithm of the IC$_{50}$ value (in molar concentration). These descriptors include topological, geometric, and electronic properties that are important for molecules to interact with biological targets. The mathematical relationship between the activity and the descriptors is represented by the MLR equation:

pIC$_{50}$ (Activity) = 2.768483965 + 0.133928895 × (Chiv5) + 1.59986423 × (bcutm1) + (-0.02309681) × (MRVSA9) + (-0.002946101) × (MRVSA6) + 0.00671218 × (PEOEVSA5) +(-0.15963415) × (GATSv4) + 0.207949857 × (J) + 0.082568569 × (Diametert).

The values of pIC$_{50}$ greater than or equal to 5.5 are likely cytotoxic, and even those near this threshold may still pose moderate toxicity risks [87]. In this study, the pIC$_{50}$ of the parent compounds IBP and KTP were 4.604 and 4.529, respectively, whereas the pIC$_{50}$ values of the degradants of IBP stayed between the limits of 4.190 and 4.603, and the degradants of KTP ranged from 4.321 to 4.705. Among the KTP, IBP, and their degradants, KTP9 showed a pIC$_{50}$ of 4.705, surpassing the parent compound, and IBP6 demonstrated the lowest predicted activity at a pIC$_{50}$ of 4.190, as illustrated in Fig 10(a) and 10(b) and Tables S6 (a) and S6 (b) in S1 File.

Table 5. The selected KTP and IBP analogs-5F19 complexes' average RMSD, RMSF, H-bond, Rg, and SASA.

| Compound | RMSD | RMSF | H-bond | Rg | SASA |
|---|---|---|---|---|---|
| KTP | 1.914±0.265 | 1.365±0.630 | 432.378±9.654 | 24.487±0.117 | 24035±322 |
| KTP2 | 1.729±0.242 | 1.272±0.569 | 432.030±10.870 | 24.486±0.159 | 24050±329 |
| KTP9 | 1.833±0.269 | 1.351±0.592 | 435.030±9.488 | 24.436±0.107 | 24840±292 |
| IBP | 1.787±0.230 | 1.315±0.599 | 430.913±10.128 | 24.515±0.135 | 24229±346 |
| IBP3 | 2.099±0.363 | 1.319±0.621 | 435.397±10.096 | 24.542±0.120 | 24047±365 |
| IBP5 | 1.952±0.319 | 1.323±0.550 | 428.496±11.543 | 24.583±0.131 | 24199±305 |

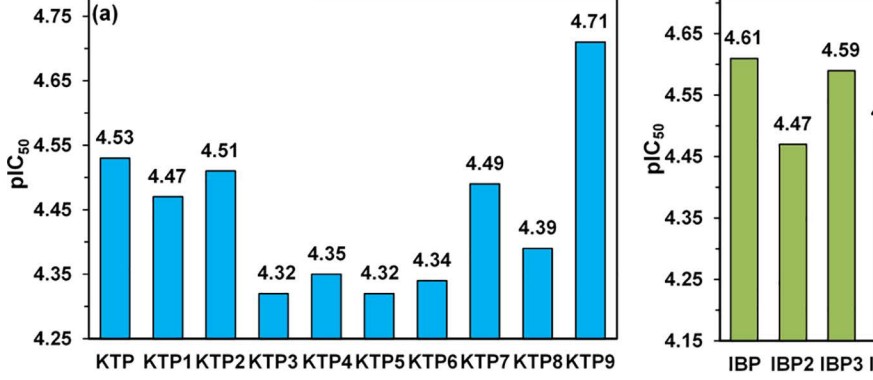

Fig 10. pIC50 studies of (a) KTP, (b) IBP, and their degradants.

## 4. Conclusion

This investigation offers a thorough in silico evaluation of Ketoprofen, Ibuprofen, and their principal degradants, demonstrating that degradation does not ensure pharmacological or toxicological inactivity of these compounds. Numerous degradants maintained stable binding affinity for the COX-2 receptor and demonstrated carcinogenic, nephrotoxic, and hematotoxic characteristics. These results indicate that NSAID degradants warrant increased assessment in drug quality control, stability evaluation, and environmental risk assessment. Future research should concentrate on the experimental verification of these predicted toxicological profiles and on methodologies to mitigate the generation of potentially harmful degradants.

## Supporting information

**S1 File. Physicochemical, spectral, medicinal, and toxicological studies of Ketoprofen, Ibuprofen, and their major degradants: a quantum-chemical and in silico approach.** S1 (a, b) Fig. Degradation pathway and IUPAC names of KTP, IBP, and their selected degradants. S1 (a, b) Table. Molecular stoichiometry of KTP, IBP, and their major degradants. S2 (a, b) Fig. HOMO-LUMO energy gaps of KTP, IBP, and their major degradants. S3 (a, b) Fig. DOS plots of KTP, IBP, and their major degradants. S2 (a, b) Table. Frontier molecular orbital (FMO) and global reactivity descriptors of KTP, IBP, and their major degradants. S4 (a, b) Fig. ESP maps of KTP, IBP, and their major degradants. S5 Fig. (a) FT-IR and (b) UV–Vis spectra (normalized) of KTP and its major degradants. S6 Fig. (a) FT-IR and (b) UV–Vis spectra (normalized) of IBP and its major degradants. S3 (a, b) Table. Selected vibrational frequencies of KTP, IBP, and their major degradants. S4 (a, b) Table. UV-Vis spectral data of KTP, IBP, and their major degradants. S5 (a, b) Table. Binding affinity and interactions of KTP, IBP, and their major degradants with the receptor protein (5F19). S7 (a, b) Fig. Nonbonding interactions and hydrogen bond surface area of IBP, and some of its degradants, with the receptor protein 5F19. S8 Fig. MD simulations over 100 ns: Radius of gyration (Rg) (a) KTP, (b) IBP; Solvent accessible surface area (SASA) (c) KTP, (d) IBP; each with KTP, IBP, and their selected degradant–5F19 complexes. S6 (a, b) Table. QSAR analysis of KTP, IBP, and their major degradants.
(DOCX)

## Acknowledgments

We sincerely thank the Computer in Chemistry and Medicine Laboratory, Dhaka, Bangladesh, for their invaluable support, guidance, and mentorship.

## Author contributions

**Conceptualization:** Monir Uzzaman.

**Data curation:** Monir Uzzaman.

**Formal analysis:** Protyoi Chakraborty, Saithajit Mohajan, Omme Samia, Nusrat Jahan, Nazmul Islam, Monir Uzzaman.

**Investigation:** Protyoi Chakraborty, Omme Samia, Monir Uzzaman.

**Methodology:** Saithajit Mohajan, Monir Uzzaman.

**Project administration:** Monir Uzzaman.

**Supervision:** Monir Uzzaman.

**Validation:** Monir Uzzaman.

**Visualization:** Protyoi Chakraborty, Saithajit Mohajan, Monir Uzzaman.

**Writing – original draft:** Protyoi Chakraborty, Saithajit Mohajan, Omme Samia, Nusrat Jahan, Nazmul Islam, Monir Uzzaman.

**Writing – review & editing:** Mahbub Alam, Monir Uzzaman.

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
