## [Decision Letter · Decision Letter 0]

7 Jan 2026

PONE-D-25-45718Physicochemical, spectral, medicinal, and toxicological studies of ketoprofen, ibuprofen, and their major degradants: A quantum-chemical and in silico approachPLOS One

Dear Dr. Chakraborty,

Thank you for submitting your manuscript to PLOS ONE. After careful consideration, we feel that it has merit but does not fully meet PLOS ONE’s publication criteria as it currently stands. Therefore, we invite you to submit a revised version of the manuscript that addresses the points raised during the review process.

We look forward to receiving your revised manuscript.

Kind regards,

Sana Shamim

Academic Editor

PLOS One

Journal Requirements:

2. Please include a separate caption for each figure in your manuscript.

Additional Editor Comments:

1.Introduction needs revision focusing on the degradation product biological response

2. In Abstract details regarding binding affinity and HOMO/LUMO values are missing

3. the article lacks the method of degradation to conduct the FTIR

4. mention x, y z coordinates

5. justify the selection of PDB Id?

6. Improve the visualization of the docking poses.

7. Conclusion needs improvement

Reviewers' comments:

Reviewer's Responses to Questions

**Comments to the Author**

1. Is the manuscript technically sound, and do the data support the conclusions?

Reviewer #1: Partly

2. Has the statistical analysis been performed appropriately and rigorously? 

Reviewer #1: N/A

3. Have the authors made all data underlying the findings in their manuscript fully available?

Reviewer #1: Yes

4. Is the manuscript presented in an intelligible fashion and written in standard English?

Reviewer #1: No

5. Review Comments to the Author

Reviewer #1: 1) what is the source of degradants? The source and origin of the the degradants have not been clearly described

2) Introduction requires a stronger scientific rationale with clearly stated objectives and and a brief outline of the findings of the study.

3) The FTIR, UV and IC50 information was provided in the result section however their experimental procedure were not given in the methodology section.

4) Conclusion section was not clear and it lacks to establish the study's implication for future use or research.

6. PLOS authors have the option to publish the peer review history of their article (what does this mean?). If published, this will include your full peer review and any attached files.

Reviewer #1: No

You may also use PLOS’s free figure tool, NAAS, to help you prepare publication quality figures: https://journals.plos.org/plosone/s/figures#loc-tools-for-figure-preparation

---

## [Author Response · Author response to Decision Letter 1]

13 Mar 2026

We thank the editor and reviewer for their valuable suggestions and corrections. We have carefully addressed all their comments and we are at their disposal for any further information.

---

## [Decision Letter · Decision Letter 1]

26 Mar 2026

PONE-D-25-45718R1Physicochemical, spectral, medicinal, and toxicological studies of Ketoprofen, Ibuprofen, and their major degradants: a quantum-chemical and in silico approachPLOS One

Dear Dr. Uzzaman,

Thank you for submitting your manuscript to PLOS ONE. After careful consideration, we feel that it has merit but does not fully meet PLOS ONE’s publication criteria as it currently stands. Therefore, we invite you to submit a revised version of the manuscript that addresses the points raised during the review process.

We look forward to receiving your revised manuscript.

Kind regards,

Sana Shamim

Academic Editor

PLOS One

Journal Requirements:

Reviewers' comments:

Reviewer's Responses to Questions

**Comments to the Author**

1. If the authors have adequately addressed your comments raised in a previous round of review and you feel that this manuscript is now acceptable for publication, you may indicate that here to bypass the “Comments to the Author” section, enter your conflict of interest statement in the “Confidential to Editor” section, and submit your "Accept" recommendation.

Reviewer #1: All comments have been addressed

Reviewer #2: (No Response)

2. Is the manuscript technically sound, and do the data support the conclusions?

Reviewer #1: Partly

Reviewer #2: (No Response)

3. Has the statistical analysis been performed appropriately and rigorously? 

Reviewer #1: N/A

Reviewer #2: (No Response)

4. Have the authors made all data underlying the findings in their manuscript fully available?

Reviewer #1: Yes

Reviewer #2: (No Response)

5. Is the manuscript presented in an intelligible fashion and written in standard English?

Reviewer #1: Yes

Reviewer #2: (No Response)

6. Review Comments to the Author

Reviewer #1: (No Response)

Reviewer #2: •Abstract is too lengthy; it should be revise to make it simple and summarize up to 300 words highlighting main outcomes of the study only!

•Very old references of year 2010 and earlier (even earlier of 1990s) are there; they should be replaced with recent references.

•It is better to mention the names of specified NSAIDs in conclusion as mentioned in title of manuscript! Further, what you have mentioned as "conclusion" may be shifted to result and discussion part while rewrite conclusion in simple 4-5 lines with final outcome of this study and future recommendations only!

•A boil egg diagram will better explain kinetic details of compounds specially for BBB and GI Absorption profiling!

•In figures 8, proper labelling of figures 8c-8f is missing! while figure 7 and figure 8 must needs improved resolutions.

7. PLOS authors have the option to publish the peer review history of their article (what does this mean?). If published, this will include your full peer review and any attached files.

Reviewer #1: No

Reviewer #2: No

---

## [Author Response · Author response to Decision Letter 2]

15 Apr 2026

We are thankful to the honorable reviewers for their valuable comments. We went through the comments, corrected accordingly, and highlighted in yellow.

•Abstract is too lengthy; it should be revise to make it simple and summarize up to 300 words highlighting main outcomes of the study only!

Response: Corrected accordingly in the Abstract section.

•Very old references of year 2010 and earlier (even earlier of 1990s) are there; they should be replaced with recent references.

Response: We have included recent and relevant references as suggested.

•It is better to mention the names of specified NSAIDs in conclusion as mentioned in title of manuscript! Further, what you have mentioned as "conclusion" may be shifted to result and discussion part while rewrite conclusion in simple 4-5 lines with final outcome of this study and future recommendations only!

Response: Corrected accordingly in the Conclusion section.

•A boil egg diagram will better explain kinetic details of compounds specially for BBB and GI Absorption profiling!

Response: Thanks to the reviewer for this suggestion. We have included boil egg diagrams in Figs 8 (a) and (b). We have also explained them under the section 3.8.

•In figures 8, proper labelling of figures 8c-8f is missing! while figure 7 and figure 8 must needs improved resolutions.

Response: Corrected accordingly. However, figure 8 has been replaced and the current figure number for fig 8 is fig 9.

---

## [Decision Letter · Decision Letter 2]

20 Apr 2026

Physicochemical, spectral, medicinal, and toxicological studies of Ketoprofen, Ibuprofen, and their major degradants: a quantum-chemical and in silico approach

PONE-D-25-45718R2

Dear Dr. Monir Uzzaman,

We’re pleased to inform you that your manuscript has been judged scientifically suitable for publication and will be formally accepted for publication once it meets all outstanding technical requirements.

Kind regards,

Sana Shamim

Academic Editor

PLOS One

Additional Editor Comments (optional):

Reviewers' comments:

Reviewer's Responses to Questions

**Comments to the Author**

1. If the authors have adequately addressed your comments raised in a previous round of review and you feel that this manuscript is now acceptable for publication, you may indicate that here to bypass the “Comments to the Author” section, enter your conflict of interest statement in the “Confidential to Editor” section, and submit your "Accept" recommendation.

Reviewer #2: (No Response)

2. Is the manuscript technically sound, and do the data support the conclusions?

Reviewer #2: (No Response)

3. Has the statistical analysis been performed appropriately and rigorously? 

Reviewer #2: (No Response)

4. Have the authors made all data underlying the findings in their manuscript fully available?

Reviewer #2: (No Response)

5. Is the manuscript presented in an intelligible fashion and written in standard English?

Reviewer #2: (No Response)

6. Review Comments to the Author

Reviewer #2: As authors incorporated are required suggestions in the revised manuscript so now it can be accepted for publication!

7. PLOS authors have the option to publish the peer review history of their article (what does this mean?). If published, this will include your full peer review and any attached files.

Reviewer #2: No

---

## [Editor Report · Acceptance letter]

PONE-D-25-45718R2

PLOS One

Dear Dr. Uzzaman,

I'm pleased to inform you that your manuscript has been deemed suitable for publication in PLOS One. Congratulations! Your manuscript is now being handed over to our production team.

Kind regards,

on behalf of

Dr. Sana Shamim

Academic Editor

PLOS One